# *MET* Amplification in Non-Small Cell Lung Cancer (NSCLC)—A Consecutive Evaluation Using Next-Generation Sequencing (NGS) in a Real-World Setting

**DOI:** 10.3390/cancers13195023

**Published:** 2021-10-07

**Authors:** Christoph Schubart, Robert Stöhr, Lars Tögel, Florian Fuchs, Horia Sirbu, Gerhard Seitz, Ruth Seggewiss-Bernhardt, Rumo Leistner, William Sterlacci, Michael Vieth, Christoph Seidl, Michael Mugler, Markus Kapp, Wolfgang Hohenforst-Schmidt, Arndt Hartmann, Florian Haller, Ramona Erber

**Affiliations:** 1Institute of Pathology, University Hospital Erlangen, Friedrich-Alexander-Universität Erlangen-Nürnberg (FAU), 91054 Erlangen, Germany; robert.stoehr@uk-erlangen.de (R.S.); lars.toegel@uk-erlangen.de (L.T.); arndt.hartmann@uk-erlangen.de (A.H.); florian.haller@uk-erlangen.de (F.H.); ramona.erber@uk-erlangen.de (R.E.); 2Comprehensive Cancer Center Erlangen-EMN (CCC ER-EMN), 91054 Erlangen, Germany; florian.fuchs@uk-erlangen.de (F.F.); Horia.Sirbu@uk-erlangen.de (H.S.); 3Department of Medicine 1, University Hospital Erlangen, Friedrich-Alexander-Universität Erlangen-Nürnberg (FAU), 91054 Erlangen, Germany; 4Department of Thoracic Surgery, University Hospital Erlangen, Friedrich-Alexander-Universität Erlangen-Nürnberg (FAU), 91054 Erlangen, Germany; 5Institute of Pathology, Neuropathology, Molecular Diagnostics and Cytology, Klinikum Bamberg, Sozialstiftung Bamberg, 96049 Bamberg, Germany; gerhard.seitz@sozialstiftung-bamberg.de; 6Department of Medicine 5, Klinikum Bamberg, Sozialstiftung Bamberg, 96049 Bamberg, Germany; ruth.seggewiss-bernhardt@sozialstiftung-bamberg.de; 7Department of Medicine 4, Klinikum Bamberg, Sozialstiftung Bamberg, 96049 Bamberg, Germany; rumo.leistner@sozialstiftung-bamberg.de; 8Institute of Pathology, Friedrich-Alexander-Universität Erlangen-Nürnberg (FAU), Klinikum Bayreuth, 95445 Bayreuth, Germany; william.sterlacci@klinikum-bayreuth.de (W.S.); michael.vieth@klinikum-bayreuth.de (M.V.); 9Diagnosticum, Pathology and Cytology, 95032 Hof, Germany; seidl@diagnosticum.eu (C.S.); mugler@diagnosticum.eu (M.M.); 10Diagnosticum, Laboratory Medicine, Microbiology, Pathology, Human Genetics, 09221 Neukirchen, Germany; 11Department of Gastroenterology, Hepatology and Infectiology, Section Hematology & Oncology, Sana Klinikum Hof, 95032 Hof, Germany; Markus.Kapp@Sana.de; 12Department of Cardiology, Sana Klinikum Hof, 95032 Hof, Germany; Wolfgang.Hohenforst-Schmidt@Sana.de

**Keywords:** NSCLC, next-generation sequencing, *MET*, fluorescence in situ hybridization, amplification, precision medicine, MET inhibitor, resistance mechanism

## Abstract

**Simple Summary:**

Lung cancer has a high incidence and affects both men and women. Targeted therapy options directed at certain mutant proteins, and which avoid systemic chemotherapy are already available and emerging. The gene mesenchymal epithelial transition (*MET*), encoding a receptor tyrosine kinase protein, is amplified in a subpopulation of lung cancer patients. The aim of our consecutive study was to assess whether next-generation sequencing (NGS) is a reliable method for the detection of *MET* gene copy number. Our study confirmed that NGS is able to detect cases harboring a high-level *MET* gene amplification but is unreliable and fails to detect the various levels of *MET* gene amplification. Therefore, NGS cannot replace the gold standard method of fluorescence in situ hybridization for the detection of *MET* gene copy number.

**Abstract:**

In non-small cell lung cancer (NSCLC), approximately 1–3% of cases harbor an increased gene copy number (GCN) of the *MET* gene. This alteration can be due to de novo amplification of the *MET* gene or can represent a secondary resistance mechanism in response to targeted therapies. To date, the gold standard method to evaluate the GCN of *MET* is fluorescence in situ hybridization (FISH). However, next-generation sequencing (NGS) is becoming more relevant to optimize therapy by revealing the mutational profile of each NSCLC. Using evaluable *n* = 205 NSCLC cases of a consecutive cohort, this study addressed the question of whether an amplicon based NGS assay can completely replace the FISH method regarding the classification of *MET* GCN status. Out of the 205 evaluable cases, only *n* = 9 cases (43.7%) of *n* = 16 high-level *MET* amplified cases assessed by FISH were classified as amplified by NGS. Cases harboring a *MET* GCN > 10 showed the best concordance when comparing FISH versus NGS (80%). This study confirms that an amplicon-based NGS assessment of the *MET* GCN detects high-level *MET* amplified cases harboring a *MET* GCN > 10 but fails to detect the various facets of *MET* gene amplification in the context of a therapy-induced resistance mechanism.

## 1. Introduction

In solid tumors, driver mutations frequently occur in receptor tyrosine kinases (RTKs) [1]. The proto-oncogene *MET* (mesenchymal epithelial transition, c-MET or hepatocyte growth factor receptor, HGFR) encodes an RTK protein, which is mainly expressed on the surface of epithelial cells and is physiologically activated by the binding of the HGF ligand (summarized in [2,3]). Binding of the ligand induces tyrosine phosphorylation, leading to homodimerization of the MET receptor. The signaling cascade leads to a change in gene expression, causing cell cycle progression, cell proliferation and increased motility and invasion [2].

In a variety of malignant tumors, the *MET* gene represents one of the drivers of tumorigenesis due to genetic aberrations, including germline or somatic point mutations, splice-site mutations leading to the skipping of exon 14 (*MET* exon 14 skipping) or gene amplification [2,4].

The genetic architecture of the *MET* gene locus is organized in a ladder-like structure characterized by many inverted repeats and therefore resembles an area of common chromosomal fragile sites [5]. These sites are susceptible to genetic aberrations, including gene amplifications. Amplification of the *MET* gene mainly occurs in lung cancer patients but can also develop in adenocarcinomas of the gastroesophageal junction (3.3%) or in glioblastoma (1.7%) [6].

The necessity of *MET* GCN evaluation in advanced NSCLC is apparent since *MET*-amplified (non-squamous) NSCLCs has shown significantly poorer prognosis in previous studies [7,8]. Regarding the amplification of the *MET* gene, a high-level amplification of *MET* can be found only in a low number of TKI-naïve NSCLC patients with frequencies ranging from 2.0–4.0% [6,9,10,11]. Notably, the definition of high-level amplified *MET* varies between studies and the test method employed.

Amplification of the *MET* gene can be classified as a gain in the gene copy number (GCN), i.e., the *MET* gene is multiplied in relation to centromere 7 (*CEN7*) or in comparison to normal, healthy tissue. In addition, it can be classified as amplified due to polysomy, i.e., the *MET* gene and the centromere *CEN7* are both increased in number [12,13]. Of clinical importance, these *MET* gene aberrations can occur either de novo as an early event in tumorigenesis or as an acquired resistance mechanism due to the use of EGFR tyrosine kinase inhibitors [9,10,14,15]. More recent data also indicate that the use of next-generation ALK inhibitors induces such a resistance phenotype characterized by *MET* gene amplification [16]. The amplification itself can be targeted with multikinase or MET-specific inhibitors to improve the outcome [7,14,17]. In 2011, the first efficient response against *MET*-amplified lung cancer using a tyrosine kinase inhibitor (crizotinib) was reported in women harboring a high-level *MET* amplification, determined by FISH (MET/CEN7 ratio > 5) [17]. There are only limited data concerning intermediate- to low-level amplification of the *MET* gene and the associated efficiency of a drug response. In 2014, it was shown that a patient suffering from NSCLC harboring low-level amplification of the *MET* gene (i.e., *MET/CEN7* ≥ 1.8 ratio ≤ 2.2, *n* = 1) had no response to the tyrosine kinase inhibitor crizotinib. In contrast, patients with tumors harboring an intermediate amplification (i.e., *MET/CEN7* ratio >2.2 to <5, *n* = 6) of the *MET* gene showed a durable response [18].

Capmatinib, another MET tyrosine kinase inhibitor, showed high antitumor activity against *MET* exon 14-mutated tumors of naïve, untreated NSCLC patients. In addition, it was especially effective against high-level *MET* amplified tumors (*MET* GCN ≥ 10) compared to low-level amplified tumors [19].

In a recent study, the authors claimed that *MET* gene amplification is not always a result of treatment but can pre-exist and be selected for during TKI treatment [20]. The use of a combinatory regimen including EGFR tyrosine kinase inhibitors and a MET inhibitor could be a successful strategy in the near future if the resulting toxicity is well tolerated.

Hence, the assessment of *MET* copy number changes should be an integral part of the diagnostic workup in advanced NSCLC. For the evaluation of the *MET* amplification status, fluorescence in situ hybridization (FISH) is an adequate and well-established method that captures the various facets of *MET* gene amplification, including “true” high-level *MET* gene amplified cases characterized by a high *MET* GCN (≥6 per cell) without concomitant polysomy (i.e., a high *MET/CEN7* ratio).

According to various studies, multigene testing for nonsynonymous mutations, translocations and copy number variations of a variety of genes is recommended to identify optimal treatment options for patients with advanced tumors, including NSCLC [21,22,23]. Next-generation sequencing (NGS), which does not reveal genetic aberrations of only one but multiple genes at the same time, represents a tissue-sparing alternative for the detection of various genetic aberrations. Another advantageous aspect of the NGS approach is the detection of *MET* exon 14 skipping aberrations, which cannot be detected by FISH but is of high clinical relevance [4]. Various methods exist to define the copy number of a gene by NGS summarized in [24].

The present study utilized consecutive, real-world data to investigate whether a multiplex, PCR amplicon-based NGS-based determination of the *MET* GCN is able to replace the FISH-based approach. In addition, the present study aimed to investigate whether an NGS-based determination of the *MET* GCN status detects all the various facets of *MET* amplification.

## 2. Materials and Methods

### 2.1. Cohort

FFPE specimens of *n* = 327 consecutive NSCLC cases diagnosed at the Institute of Pathology of the University Hospital Erlangen, Germany, between July 2016 and May 2018 were included in this study. All cases were therapy-naïve except for one patient who had undergone EGFR TKI therapy (Case #3) for one year. Of a total of *n*= 327 samples, *n* = 205 samples could be analyzed by next-generation sequencing (NGS) and fluorescence in situ hybridization (FISH) for genetic aberrations, including the status of the *MET* gene copy number (GCN) (also see results Section 3.1). Cases that were not analyzed by *MET* FISH during routine diagnostics were retrospectively analyzed using *MET* FISH. The various analyses were performed double-blinded. Ethical approval was obtained by the local ethics committee (85_17B).

### 2.2. MET GCN Detection by NGS

H&E staining was performed to confirm the diagnosis of NSCLC and to estimate the tumor cell content. After microdissection of the tumor tissue, DNA was isolated using standard techniques. Regions of interest, including the *MET* gene, were enriched and amplified according to the manufacturers’ instructions, using a multiplex, PCR amplicon-based 15-gene panel (TruSight Tumor 15, TST15 Illumina, Inc., San Diego, CA, USA). In detail, the regions of interest such as the *MET* gene or the *ERBB2* gene were amplified using tagged, gene specific primers followed by target indexing. After this, libraries were cleaned up and the quality was checked via agarose gels or with the aid of a tape station. Finally, the various libraries were pooled and were run on a MiSeq, respectively (Illumina). The obtained sequences (.fastq files) were bioinformatically analyzed using the TruSight Tumor 15 application in the BaseSpace Sequence Hub (Illumina) and aligned to the reference sequence hg19. Molecular aberrations, including point mutations, deletions and insertions, were annotated and described using standard HGVS nomenclature. In addition, changes in gene copy numbers were determined by the CRAFT copy number variant caller (v1.0.0.12) algorithm incorporated in the TST15 application. This algorithm is designed for tumor samples without matched normal controls and can detect amplifications/deletions in three genes (*EGFR*, *ERBB2* and *MET*) above a 1.6-fold change.

### 2.3. MET GCN Detection by FISH

The *MET* FISH protocol, established in-house for routine diagnostics, was performed on 1–2 µm-thick freshly cut sections from FFPE NSCLC tumor blocks using a *MET* dual-color probe (ZytoLight^®^ SPEC *MET/CEN7* Dual Color Probe, Z-2087–50, ZytoVision GmbH, Bremerhaven, Germany) according to the manufacturers’ standard protocol (ZytoLight FISH-Tissue Implementation Kit, Z-2028-5, ZytoVision GmbH) and the routine in-house standards. Pepsin digestion was performed at 37 °C for 9 min, first hybridization at 75 °C for 10 min and the second step at 37 °C overnight. The *MET* Dual Color Probe used comprises one orange fluorochrome (ZyOrange)(ZytoVision GmbH, Bremerhaven, Germany) directly labeled *CEN7* probe binding specific to the alpha satellite centromeric region of chromosome 7 (D7Z1) and one green labeled (ZyGreen) (ZytoVision GmbH, Bremerhaven, Germany) probe binding the *MET* gene located at 7q31.2.

The *MET* FISH status was analyzed, blinded to NGS data, using a Leica fluorescence microscope at 1000× magnification (100 × oil objective, Leica Microsystems GmbH, Wetzlar, Germany). A DAPI filter was used to visualize nuclei, and a double-bandpass filter (green/orange) was used to quantify *MET* and *CEN7* signals. After reviewing the H&E slide of each NSCLC case and reidentifying the tumor cells in DAPI, 50 nonoverlapping tumor nuclei were evaluated, and both green and orange signals per nucleus were counted to determine the mean GCN of *MET* and *CEN7*, respectively. Cases were considered invalid if signals were hardly detectable or missing or if background staining or autofluorescence was too strong. The *MET* gene copy number status was classified into four groups according to Schildhaus et al. (high-, intermediate- and low-level amplification or normal, nonamplified) [13]. Tumor cells harboring *CEN7* signals on average ≥3.6 were classified as polysomic [25].

### 2.4. Statistical Analysis

Venn diagrams were generated using the BioVenn-Web tool [26]. The Oncoprinter illustration option of cBioPortal was used to correlate mutations according to the sex and age of the patients [27,28].

## 3. Results

### 3.1. Comparison of the MET GCN of the NSCLC Cohort Using Two Different Methods: FISH versus NGS

Out of *n* = 327 consecutive NSCLC cases in total, *n* = 107 cases could not be analyzed via FISH due to an insufficient quality of the fluorescence signal, limited amount of material or lack of the corresponding material (formalin-fixed and paraffin-embedded (FFPE) material from external pathologies). Out of the overall cohort (*n* = 327 NSCLC cases), *n* = 23 cases could not be analyzed via NGS due to a limited amount of material or due to insufficient coverage of the *MET* gene locus (minimum amplicon count of >200). Overall, *n* = 205 NSCLC cases could be directly compared using both NGS and FISH regarding the determination of the *MET* GCN status.

Figure 1 depicts the various facets of *MET* gene amplification assessed by FISH in the present study. High-level *MET* amplified cases, characterized by a high *MET/CEN7* ratio (≥2.0) or cluster formation (Figure 1a,b), and high-level *MET* amplified cases, characterized by a high *MET* GCN (≥6.0), accompanied by a polysomic state of the cells, were detected in the present study cohort (Figure 1c).

In the cohort of *n* = 205 evaluable cases, *n* = 42 cases showed an aberrant GCN of *MET* assessed by FISH (Table 1). These cases harbored various levels of *MET* gene amplification ranging from low to high levels. Of the *n* = 42 cases, *n* = 24 cases were characterized by polysomy (57.0%). Focusing on the clinically relevant high-level *MET* amplified cases assessed by FISH, *n* = 16 cases of *n* = 42 cases showed a *MET*/*CEN7* ratio ≥2.0 or an average *MET* GCN per cell ≥6.0 or ≥10% of tumor cells containing ≥15 *MET* signals (Table 1 and Figure 2a). In the cohort of *n* = 205 cases, *n* = 9 cases harbored a “true” high-level *MET* gene amplification assessed by FISH, i.e., cases without a concomitant polysomy (4.4%).

Compared to these findings, *n* = 9 cases were classified as *MET* gene amplified assessed by the NGS approach (Table 1 and Figure 2a).

Comparing both methods regarding the status of the *MET* GCN revealed a discrepancy of *n* = 35/43 cases (81.4%). Focusing on both, the high-level *MET* amplified cases assessed by FISH and the *MET* amplified cases assessed by NGS showed a better concordance. Six of *n* = 16 cases classified as high-level *MET* amplified by FISH were classified as *MET* amplified by NGS (37.5%, cases #2–6 and #15, Figure 2a). Cases harboring a *MET* GCN of >10.0 showed the best concordance between FISH and NGS.

This observation emphasizes that only cases harboring a high *MET* GCN or relatively high *MET/CEN7* ratio classified by FISH can be detected adequately by the NGS approach. However, one of five cases was not classified as *MET* gene amplified by NGS.

Furthermore, looking at the overlap of *n* = 6 cases, *n* = 2 cases showed a concomitant polysomy determined by FISH. Case #3 represents a patient with an *EGFR* mutation who had undergone EGFR TKI therapy for one year (see Material and Methods). The high-level amplification of the *MET* gene in this case likely represents a resistance mechanism as a consequence of EGFR TKI therapy.

The *n* = 10 cases that were not classified as *MET* amplified by NGS but showed a clear high-level amplification by FISH demonstrates that not only polysomic *MET* amplified samples, but also cases harboring a high *MET* GCN >10 are not reliably detected by the NGS approach (Cases #1, 7-14 and 16, Figure 2a). Case #1, yielding a *MET* GCN >10 and a *MET/CEN7* ratio >5, revealed a huge discrepancy between FISH and NGS. This case showed cluster formation of the *MET* signal assessed by FISH.

The *n* = 3 cases that were classified as *MET* gene amplified by the NGS approach but were not defined as high-level amplified by FISH harbored an intermediate-/low-level amplification, or completely lacked amplification of *MET* (Cases #30–31 and 43, Figure 2a). Case #43 was classified as having the *MET* gene amplified by NGS but showed no *MET* amplification when assessed by FISH. The mean coverage of the *MET* locus in this particular case was 11.000, whereas the mean coverage of the *CEN7* region was 27.300. With a normalized ratio of *MET/CEN7* of 0.40, calculated manually based on the amplicon coverages, the TST15-based algorithm defined this particular case as *MET* gene amplified for unknown reasons.

With regard to the normal, non-*MET* gene amplified cases, only *n* = 1 case was classified as non-*MET* amplified by FISH but determined as *MET* amplified by NGS (Case #43, Table 1 and Figure 2b). Furthermore, mainly intermediate- and low-level *MET* gene amplified cases were missed by the NGS approach (Figure 2b). Many of these cases showed a concomitant polysomy of the cells. For *n*= 162 cases, both NGS and FISH approaches consistently classified the NSCLC cases as non-*MET* gene amplified (i.e., normal).

In summary, comparing FISH versus NGS regarding the status of the *MET* GCN revealed a discrepancy of *n* = 48 of *n* = 205 cases (23.4%, Figure 2a,b).

### 3.2. Correlations of MET GCN with Patient Data

Investigation of a potential correlation of an NGS-based identification of *MET* gene amplification and the sex and/or age of the patients in the cohort revealed a trend for an enriched amplification of *MET* in males (Figure 3a). No correlation was observed between *MET* gene amplification status and patient age (Figure 3b). Interestingly, the presence of *MET* gene amplification determined by NGS did not exclude a concurrent mutation in oncogenic driver genes. Focusing on the *MET* gene itself, our cohort demonstrates that *MET* gene amplification was never accompanied by a single-nucleotide variant (SNV) in the *MET* gene itself.

Amplifications of *ERBB2* and *EGFR*, as well as somatic point mutations in the proto-oncogene *KRAS*, were observed (cases #1, 3, 17, and 19). Case #3 harbored an in-frame exon 19 deletion mutation in the *EGFR* gene. Case #1 was characterized by a high-level *MET* amplification assessed by FISH, whereas case #17 showed an intermediate-level amplification and case #19 showed a normal level of the *MET* gene. Another notable point relates to the fact that somatic SNVs in the *EGFR* gene are often accompanied by amplification of the gene itself. In contrast, amplification of the *ERBB2* or *MET* gene is usually not associated with a concomitant SNV at the DNA level.

## 4. Discussion

In this study, we directly compared *n* = 205 consecutive NSCLC cases from routine diagnostics regarding the evaluation of *MET* GCN and the type of *MET* gene copy aberration using either an amplicon-based, 15-gene NGS panel or the standard FISH method. Out of the evaluable *n* = 205 cases, *n* = 9 cases (4.4%) were classified as *MET* amplified by NGS. In contrast, *n* = 16 cases were classified as high-level *MET* amplified by FISH (including polysomic cases). Only *n* = 6 of the amplified cases, determined by NGS, were concurrently classified as *MET* amplified by FISH, yielding a discrepancy of *n* = 7 (43.7%) cases. Focusing on the cases harboring a *MET* GCN >10 as assessed by FISH, the NGS approach reliably classified 80.0% of these cases as *MET* amplified (*n* = 4/5).

Our study shows for the first time that using the small focus gene panel TST15 from Illumina one is able to detect *MET* amplified cases, but not reliably. In line with previously published data, our study showed that using an NGS approach, high-level *MET* amplified cases (*MET* GCN >10.0, [13]) can be detected, even though not reliable. Various aspects of this study again emphasized that an exclusive NGS-based determination of the *MET* GCN cannot completely replace a FISH-based assessment of the *MET* gene status. Heydt et al. who used custom based NGS panels from Qiagen or an Ion AmpliSeq Custom panel from Thermo Fisher already faced similar problems [29]. They convincingly showed that MET IHC had the best concordance with *MET* FISH when comparing *n* = 35 *MET* amplified samples (low-, intermediate- and high-level *MET* amplification). In contrast, the NanoString copy number assay, ddPCR copy number assay and custom amplicon-based parallel sequencing showed a lower concordance, especially when looking at the low-level *MET*-amplified samples. Furthermore, they showed that high-level *MET*-amplified cases generally showed better concordance between NGS and FISH than intermediate- or low-level *MET*-amplified cases. They claim that only *MET* high-level amplified samples harboring a GCN ≥6 determined by FISH can be detected via alternative methods, including NGS. Additionally, other groups such as Guo et al. and Clavé et al. already compared the specificity and sensitivity of various methods for the detection of *MET* gene amplification [30,31]. Guo et al. mainly focused on the question of whether a MET overexpression detected by IHC could be correlated to a *MET* gene amplification or *MET* exon 14 skipping. They conclude that IHC is not a useful method for the detection of genomic changes of the *MET* locus as with gene amplification or *MET* exon 14 skipping. We strictly focused on the amplification of the *MET* gene, comparing the two methods, FISH versus NGS. Moreover, two institutes performing NGS used a hybrid capture based assay, not a PCR amplicon-based one as we did in our analysis. Using a hybrid capture based NGS assay, one investigates a larger gene panel and could lose some information of certain gene loci. In contrast, amplicon sequencing has a higher on target rate yielding a high specificity and deep coverage.

In contrast to [31] their biased, non-consecutive approach selecting *n* = 26 *MET* gene amplified (FISH; *MET* GCN >5, Cappuzzo Score) samples of in total *n* = 222 NSCLC cases, we used an unbiased consecutive, double blinded approach. In addition, they used a 500+ NGS gene panel (PGDx elioTM) and set a cut off for *MET* gene amplification of >3.0-fold change. In our study, cases harboring a fold change >1.6 were classified as *MET* gene amplified by NGS. For assessment of *MET* amplification by FISH, they referred to two individual FISH score references (Cappuzzo score and UCCC FISH-criteria). In our study, we referred to only one FISH scoring system [13].

In our hands, the PCR amplicon based NGS approach was able to detect cases harboring a high *MET* GCN >10, even in the presence of low polysomy (*n* = 2 cases). However, the NGS approach was not able to detect the various facets of *MET* amplification. No certain log fold changes could be defined that would segregate intermediate- to low-level *MET* amplification. Only *n* = 2 cases (Case #30, 31) harboring an intermediate- or low-level *MET* amplification assessed by FISH were also classified as amplified by NGS. In most cases, the NGS approach classified *MET* GCN as normal, although the corresponding FISH samples yielded intermediate- to low-level *MET* gene amplification.

Whether a discrimination between “true” high-level *MET* gene amplified cases and intermediate- to low-level *MET* amplified cases is of clinical relevance is still debated. “True” high-level *MET*-amplified cells are characterized by a high MET/CEN7 ratio without concomitant polysomy. In a retrospective study, it was investigated whether “true” high-level *MET* amplified cases responded more efficiently to targeted therapy than intermediate- to low-level *MET* amplified cases [32]. The study highlighted that only “true” high-level *MET* amplified cases harboring MET/CEN7 ratios ≥5.0 efficiently responded to targeted therapy.

The discrepancy between the FISH and NGS approaches observed in the present study could be due to biological or technical reasons regarding the *MET* gene locus. For example, the mean coverage of the *ERBB2* gene in the present study comprised 15.000 reads, whereas the average coverage of the *MET* gene comprised a far lower amount of less than 5.000 reads (data not shown). Although amplification of the *MET* gene takes place, the number of amplified reads could still be low and could impede PCR amplification and subsequent detection.

Another issue concerns the tumor content and tumor purity of a sample. If the tumor content is too low, NGS might show false negative results, as the normal cell content could distort the results. Therefore, carefully concerted microdissection of the tumor area is crucial to avoid contamination with normal tissue [33].

From a technical point of view, the way the *MET* gene itself is organized on the chromosome could be problematic. The *ladder-like* structure characterized by many inverted repeats makes the gene locus prone to gene amplification but may also complicate sophisticated primer design [5]. Perhaps the primer-annealing efficiency in these regions is impaired, leading to lower coverage than expected. In addition, a high content of the two nucleotides guanine and cytosine (GC content) of the *MET* locus could impair appropriate dissociation of the DNA strands during the PCR denaturation step.

Our study also highlights the limitations of the NGS approach with regard to the polysomic state of the cells. For example, cases #7, 8 and 10–12 showed a high-level amplification of the *MET* gene assessed by FISH, whereas the NGS method classified these cases as normal. These *n* = 5 cases were all polysomic, thus not resembling the “true” high-level *MET* amplified samples. Looking at the coverage of the corresponding NGS amplicon files revealed high coverage values of the *CEN7* control region and lower coverage values for the corresponding *MET* locus. This observation could suggest that the TST15 CRAFT algorithm missed these high-level *MET* amplified cases, probably due to a defined cut-off, which depends on the calculation of the median normalized bin count of the target gene versus the median bin count of the entire panel. In the case of a polysomic state of the cell, *MET* amplification could be masked.

Using the corresponding normal healthy tissue of a tumor sample as a reference for the calculation of the ratios and a normal reference pool could circumvent such a problem. Grasso et al. used a PCR amplicon-based approach and developed an algorithm for the assessment of GCN alterations that used a sequence data pool of normal samples as a reference [33]. They convincingly showed that they detected clinically relevant GCN alterations (i.e., *ERBB2*, *EGFR*, *MET*) for *n* = 14 breast cancer samples using the matched normal samples or a normal reference pool. Similarly, Niu et al. defined an *ERBB2* breast cancer-specific cutoff for the NGS algorithm by sequencing *n* = 151 *ERBB2* nonamplified FFPE samples [34]. By this approach, they also created a normal reference pool. With the aid of using a normal reference pool, the sensitivity of the NGS GCN determination could be increased, as a polysomic state of the cells could not mask the detection of a possible GCN alteration. Moreover, one could save money by avoiding sequencing the matching normal tissue of a tumor specimen.

Another critical disadvantage for the assessment of *MET* amplification by NGS is that cells harboring cluster formation of the *MET* gene locus can be sparse and therefore can be masked by surrounding tumor cells without cluster formation (see case #1, Table 1). The NGS-based algorithm might miss such cells and, hence, would define such samples as false negatives.

Another limitation of the TST15 NGS approach is that the customer does not receive any detailed information on ratios or total gene copy numbers. Although the newer gene panels TST170 or TSO500 (both Illumina) make use of the same CRAFT gene copy number algorithm, these new methods offer the customer values of fold changes and a possibility to determine the total gene copy number of a gene of interest. Furthermore, other technologies, such as the FoundationOne Assay, used in the study of Frampton et al., provide the customer with total gene copy number values. In this particular case, they defined samples harboring ≥6.0 gene copies of the *MET* gene as amplified [23].

Our study also focused on a possible correlation between the sex and age of the patients and concomitant *MET* gene amplification determined by NGS. The trend of more males having an amplification of the *MET* gene is in line with data from Okuda et al., showing that in a cohort of *n* = 213 NSCLC cases (*n* = 148 males and *n* = 65 females), all *MET* amplified cases were males [35]. Although these males were all smokers, hindering a clear correlation between sex and *MET* gene amplification, smoking behavior could also lead to the observed phenotype. Information concerning female smokers is lacking in this particular case.

Another interesting issue, which was addressed by the present study, was the occurrence of concomitant driver gene mutations in the presence of *MET* amplification. In our study, *n* = 4 out of the *n* = 9 *MET* gene amplified cases determined by NGS showed a concomitant molecular alteration in the genes *KRAS*, *EGFR* and *ERBB2*. Focusing on the “true” high-level *MET* amplified cases classified by FISH, *n* = 6 of *n* = 9 cases showed a concurrent mutation of other oncogenic driver genes. These results are against the common consensus in the literature that a “true” high-level *MET* amplification is not simultaneously accompanied by a mutation in one of the known driver genes. The study of Noonan et al. revealed that the “true” high-level *MET* amplified cases explicitly did not show a concomitant mutation in another oncogenic driver gene, such as *KRAS* or *EGFR* [32]. Looking in more detail, case #15 of our study was classified as high-level *MET* amplified by the FISH approach but revealed only a *MET* GCN of 4.4 accompanied by a GCN of *CEN7* of 1.6 leading to a ratio of 2.7. Based on the classification system according to [13], this case is classified as high-level *MET* amplified but probably does not reflect a “true” high-level MET amplified case.

Regarding the MET gene itself, our cohort did not include cases that showed a concurrent mutation of the *MET* gene in the presence of *MET* gene amplification. An investigation of a cohort of *n* = 178 NSCLC cases by NGS showed that the *MET* gene itself is rarely mutated, and only *n* = 3 cases showed a somatic exon 14 deleting splice-site mutation [35]. Currently, there are no data that directly address the question of whether the amplification of the *MET* gene impedes a concurrent mutation of the gene itself.

Our rate of clinically relevant “true” high-level *MET* amplification (4.4%) using FISH is similar to the observed rate in the study by Okuda et al., where they obtained a rate of 5,6% *MET* gene amplified cases of *n* = 213 NSCLC patients [35]. Tsuta et al. described more divergent findings compared to our study cohort reporting amplification of the *MET* gene in 10,9% of NSCLC cases assessed by BISH (bright-field in situ hybridization, BISH-positive, when one of five criteria were met, for example: *MET* to *CEN7* ratio ≥2.0 or >15.0 copies of the *MET* signal in >10% of tumor cells) [36]. On the other hand, when comparing our study with the results obtained by Park et al., our rate of *MET* gene amplification was higher compared to their results showing *MET* amplification in 2.4% of *n* = 380 NSCLC cases determined by FISH using the University of Colorado Cancer Center criteria (UCCCC) [37]. Using the Capuzzo scoring system, the authors obtained a rate of 7.1% *MET* gene amplified cases. Using MET IHC, they found that 13.7% of patients showed MET overexpression. In another study that investigated a large cohort of patients with surgically resected NSCLC for MET overexpression and gene amplification, Sterlacci et al. found *MET* amplification in 2.4% of cases assessed by FISH using tissue microarrays (using a defined cut-off for amplifications of *MET* to *CEN7* signal ratio of ≥2.0) [38]. Schildhaus et al. described 3% of NSCLC being clear-cut *MET* high-level amplified. However, low- and intermediate-level amplifications were assessed in 30% of investigated NSCLCs [13]. In summary, these data show that the comparability between different laboratories is difficult and that the interpretation of *MET* amplification regarding the prognostic value and the various levels of *MET* amplification, respectively, is challenging.

The disadvantages of our study comprise the partly retrospective character of the study (assessment of *MET* FISH). Moreover, we were not able to directly link the *MET* GCN alteration status with smoking behavior and prognosis of the respective NSCLC patients. In addition, we were not able to monitor whether these patients obtained targeted therapy. One reason for this lack of information is that some cases had been sent from external pathologies. Another point of criticism is the fact that 32.7% of our cohort could not be evaluated using *MET* FISH due to insufficient quality of the fluorescence signal, limited amount of material, or lack of the corresponding material. These limitations were also encountered in other studies where for example silver in-situ hybridization (SISH)--- was not assessable in 15% of patients [39]. Regarding NGS, data were evaluable in 93% of our cohort. Only 7% of the samples could not be analyzed via NGS due to insufficient amount of the material or low coverage of the *MET* gene locus. This highlights a positive aspect of the NGS method as—although material can be sparse—the PCR amplification step and the highly sensitive sequencing platform make it possible to analyze a specimen, which could be potentially problematic for FISH.

In summary, our data agrees with previous studies in that a “true” high-level *MET* amplification is still a rare event in NSCLC tumorigenesis. In the near future, however, it could become more prominent as a potential secondary resistance mutation in response to an increasing number of prescribed tyrosine kinase inhibitor therapies. Therefore, the need for a harmonized scoring system using FISH, which enables a comparison between laboratories and the dissection of high-, intermediate- and low-level amplified cases, is still crucial. In addition, one should be aware of the limitations that arise when *MET* GCNs or GNCs in general are assessed by NGS. The use of normal reference pools in the case of *MET* GCN determination should be carefully considered, as nonclinical relevant high-level polysomic cases could also be included in this way of therapy stratification. Of interest, TKI-induced, resistance-associated amplification of the *MET* gene is not exclusively characterized by high-level *MET* amplification, as intermediate- to low-level amplifications are also observed [16]. Therefore, FISH-based determination of *MET* GCN is currently inevitably necessary to detect the various facets of *MET* gene amplifications.

## 5. Conclusions

The findings of the present study are of great importance, as the future of routine molecular diagnostics will be mainly based on NGS data. As gene amplifications can be obtained easily by NGS, one would be tempted to avoid a time- and tissue-consuming additional method such as FISH. However, NGS does not resemble a reliable substitute for the assessment of *MET* gene copy number. Using FISH during routine diagnostics for the assessment of the *MET* gene copy number will remain necessary to reliably detect the various levels of *MET* gene amplification.

## Figures and Tables

**Figure 1 cancers-13-05023-f001:**
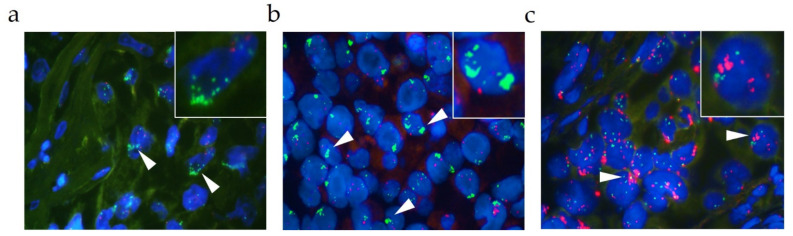
Overview of the various facets of a *MET* amplification assessed by FISH (each 1000× magnification). Representative fluorescence images showing a high-level *MET* gene amplification characterized by (**a**) a high *MET* GCN, (**b**) cluster- formation (**c**) or accompanied polysomy. Green signals represent the *MET* probe, red signals represent the CEN7 probe. White arrows indicate the relevant areas. Images in the upper right are magnified areas of the analyzed slide.

**Figure 2 cancers-13-05023-f002:**
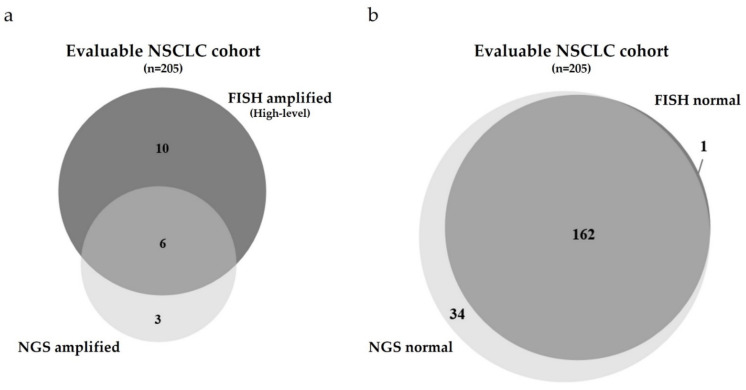
Comparison of the *MET* GCN of the NSCLC cohort using two different methods: FISH versus NGS. (**a**) Quantitative Venn Diagram of the evaluable NSCLC cases harboring a high-level amplification of the *MET* gene determined by either FISH (dark gray, polysomic cases included) or NGS (light gray, status according to the CRAFT algorithm). Overlaping cases are depicted in mid-gray color. (**b**) Quantitative Venn Diagram of the evaluable NSCLC cases showing no amplification of the *MET* gene determined by either FISH (dark gray) or NGS (light gray, status according to the CRAFT algorithm). Overlaping cases are depicted in mid-gray color.

**Figure 3 cancers-13-05023-f003:**
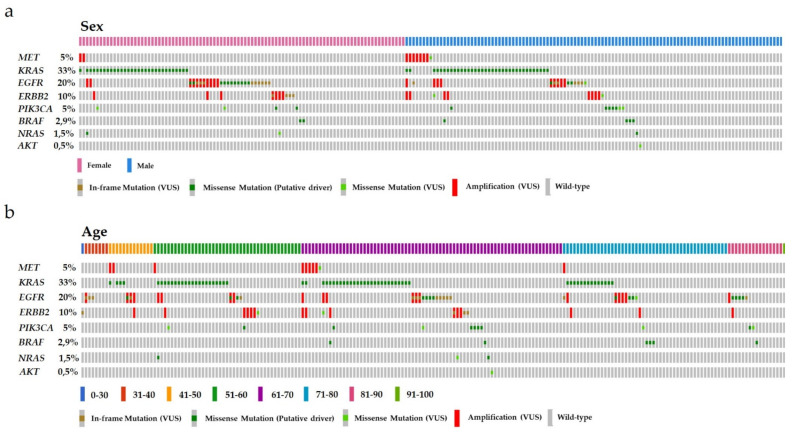
Mutational profile of the consecutive NSCLC cohort regarding the sex and age of the patients. (**a**) Arrangement of the cases according to the sex of the patients. Each column represents one case. Pink color resembles females, whereas the blue color resembles males. (**b**) Arrangement of the cases according to the age of the patients. Each column represents one case. The various age groups are colored differently (dark blue = 0–30, red = 31–40, orange = 41–50, dark green = 51–60, violet = 61–70, light blue = 71–80, pink = 81–90, light green = 91–100). In both (**a**,**b**) the affected genes are listed on the left with the frequency in the cohort shown as percent. Furthermore, in both (**a**,**b**), in-frame or missense mutations classified as variants of unknown significance (VUS) are labeled in brown and light green, respectively. Known missense mutations (putative driver mutations) are labeled in dark green. Cases harboring an alteration in gene copy numbers are depicted in red. Nonmutant, wild type samples are labeled in gray.

**Table 1 cancers-13-05023-t001:** Overview of the *MET* amplified cases assessed by FISH and NGS. Table showing cases #1–43 harboring an amplification of the *MET* gene determined by FISH or NGS. Cases are arranged according to their FISH classification, descending from high-level amplified cases to low-level amplified cases and descending from cases with the highest *MET* GCN to the lowest. The columns show the various FISH parameters (*MET*/*CEN7* ratio, *MET* GCN, *CEN7* GCN, *MET* status and polysomy) as well as the NGS status. Cases which were classified as amplified by the NGS approach are highlighted in green.

	FISH					NGS
Case-ID	*MET/CEN7* Ratio	*MET* GCN	*CEN7* GCN	*MET* Status	Polysomy	NGS Status
#1	>5.00	>10.00	2.00	High-level amplification	No	Normal
#2	>2.50	>10.0	3.00	High-level amplification	No	Amplified
#3	3.30	>10.0	3.00	High-level amplification	No	Amplified
#4	>2.50	>10.0	4.00	High-level amplification	Yes	Amplified
#5	>2.50	>10.0	4.00	High-level amplification	Yes	Amplified
#6	>4.00	>8.00	2.00	High-level amplification	No	Amplified
#7	1.10	7.60	7.20	High-level amplification	Yes	Normal
#8	1.40	7.00	5.00	High-level amplification	Yes	Normal
#9	2.00	6.00	3.00	High-level amplification	No	Normal
#10	1.32	6.00	5.00	High-level amplification	Yes	Normal
#11	1.50	6.00	4.00	High-level amplification	Yes	Normal
#12	1.30	6.00	4.50	High-level amplification	Yes	Normal
#13	2.50	5.00	2.00	High-level amplification	No	Normal
#14	3.00	4.50	1.50	High-level amplification	No	Normal
#15	2.70	4.40	1.60	High-level amplification	No	Amplified
#16	2.00	4.00	2.00	High-level amplification	No	Normal
#17	1.38	5.50	4.00	Intermediate-level amplification	Yes	Normal
#18	1.70	5.00	3.00	Intermediate-level amplification	No	Normal
#19	1.70	5.00	3.00	Intermediate-level amplification	No	Normal
#20	1.25	5.00	4.00	Intermediate-level amplification	Yes	Normal
#21	1.00	5.00	5.00	Intermediate-level amplification	Yes	Normal
#22	1.25	5.00	4.00	Intermediate-level amplification	Yes	Normal
#23	1.00	5.00	5.00	Intermediate-level amplification	Yes	Normal
#24	1.25	5.00	4.00	Intermediate-level amplification	Yes	Normal
#25	1.00	5.00	5.00	Intermediate-level amplification	Yes	Normal
#26	1.00	5.00	5.00	Intermediate-level amplification	Yes	Normal
#27	1.00	5.00	5.00	Intermediate-level amplification	Yes	Normal
#28	1.25	5.00	4.00	Intermediate-level amplification	Yes	Normal
#29	0.83	5.00	6.00	Intermediate-level amplification	Yes	Normal
#30	1.26	5.00	4.00	Intermediate-level amplification	Yes	Amplified
#31	1.50	4.90	3.20	Low-level amplification	No	Amplified
#32	1.50	4.50	3.00	Low-level amplification	No	Normal
#33	1.50	4.50	3.00	Low-level amplification	No	Normal
#34	1.50	4.50	3.00	Low-level amplification	No	Normal
#35	1.10	4.50	5.00	Low-level amplification	Yes	Normal
#36	1.13	4.50	4.00	Low-level amplification	Yes	Normal
#37	1.13	4.50	4.00	Low-level amplification	Yes	Normal
#38	1.33	4.00	3.00	Low-level amplification	No	Normal
#39	1.30	4.00	3.00	Low-level amplification	No	Normal
#40	1.14	4.00	3.50	Low-level amplification	No	Normal
#41	1.00	4.00	4.00	Low-level amplification	Yes	Normal
#42	1.00	4.00	4.00	Low-level amplification	Yes	Normal
#43	1.00	3.50	3.50	Normal	No	Amplified

## Data Availability

Data can be obtained on request. Data are not stored on publicly available servers.

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
