# Peer review of "MET Amplification in Non-Small Cell Lung Cancer (NSCLC)—A Consecutive Evaluation Using Next-Generation Sequencing (NGS) in a Real-World Setting"

_cancers, 2021, doi:10.3390/cancers13195023_

Round 1
Reviewer 1 Report
This study aims to investigate whether whether next generation sequencing (NGS) is a reliable method for the detection of the MET gene copy number. Theire conclusion is it will not replace the gold standard. I am not sure whether this findins are significant to the community. Besides, I have the following comments.
- The patient number claims in the abstract is n=327. But only n=205 cases are reported. I think the claim of n=327 is mis-leading.
- I am not sure whether the finds are significant to the research community. The authors are encourge to clarify it.
Author Response
Reviewer 1: This study aims to investigate whether whether next generation sequencing (NGS) is a reliable method for the detection of the MET gene copy number. Theire conclusion is it will not replace the gold standard. I am not sure whether this findins are significant to the community. Besides, I have the following comments.
We thank the reviewer for his/her comments on our manuscript.
The reviewer 1 mentioned that the research design is not appropriate and that the methods need to be described in more detail. Furthemore the reviewer suggested that the results should be clearly presented and the conclusions should be supported by the results.
The research design is appropriate as we investigated a consecutive, unbiased cohort of NSCLC patients. The present study represents a double-blinded analysis. We had access to many samples arriving from different pathologies. Therefore, also the various fixation methods and different tissue qualities are represented in our cohort (material and methods section 178 – 179). Of course it would have been interesting whether we would obtain the same results using an alternative NGS sequencing approach like hybrid capture technology. However, this was not possible to perform due to the limited tissue (mostly small biospies). Furthermore, it would have been interesting whether we can detect therapy induced MET amplifications by using more patients which already obtained a TKI therapy.
We added a more detailed description of the TST15 sequencing protocol (material and methods section line 187 - 191). Furthermore, we adjusted the n=327 sample number to the evaluable n= 205 samples of the cohort and referred to the results section for further explanation of a reduced total sample number (material and methods section line 174 – 177).
We think the results are clearly presented. First, we explain the aberrant total number of samples and the number of samples which could be evaluated by both NGS and FISH. Then we show examples for the various facettes of a MET gene amplification observed during the FISH analysis. The status of the MET gene locus assessed by the respective methods resembles the core of the analysis. In the last section of the results, we compared the MET gene copy number with patients´ data.
We revised the conclusions part (conclusions section line 672 – 678).
Major:
- The patient number claims in the abstract is n=327. But only n=205 cases are reported. I think the claim of n=327 is mis-leading.
We agree that mentioning in the abstract a number of n=327 NSCLC cases which were used in the study is misleading as in the results part only the evaluable n=205 cases were included. We corrected this issue in the abstract section in the revised manuscript (abstract section line 48 - 50).
- I am not sure whether the finds are significant to the research community. The authors are encourge to clarify it.
The findings of the present study are of great importance regarding MET diagnosis as also stated by reviewer 2. The future of routine molecular diagnostics will be mainly based on NGS data. As gene amplifications can be obtained easily by NGS one would be attempted to avoid a time- and tissue-consuming additional method like FISH. But our data indicate that a FISH based determination of the MET gene copy number is still inevitable. The various levels of MET gene amplification leading to different responses to MET TKI therapy need to be classified in detail. NGS is not able to dissect the various levels of MET gene amplification and still misses some high-level MET amplified cases. But the determination of a MET gene amplification is also of importance as it can also reflect a resistance mechanism in response to various targeted therapies like ALK- or EGFR-tyrosine kinase inhibitors. We emphasized this issue in the conclusions section in the revised manuscript (conclusions section line 672 – 678).

Reviewer 2 Report
Schubart et al discuss an important problem of MET diagnosis. However,
they need to clearly define the novelty of their findings and how their study differ from previous observations such as Guo et al 2019(doi: 10.1016/j.jtho.2019.06.009 and doi.org/10.1093/annonc/mdz269.009).
Additionally the authors need to correct the references.
Author Response
Reviewer 2: Schubart et al discuss an important problem of MET diagnosis. However, they need to clearly define the novelty of their findings and how their study differ from previous observations such as Guo et al 2019(doi: 10.1016/j.jtho.2019.06.009 and doi.org/10.1093/annonc/mdz269.009). Additionally the authors need to correct the references.
We thank the reviewer for his/her generally very positive evaluation of our manuscript.
- However, they need to clearly define the novelty of their findings and how their study differ from previous observations such as [1] and [2].
Our study shows for the first time that using the small focus gene panel TST15 from Illumina one faces similar problems as for example observed by Heydt et al. who used custom based NGS panels from Qiagen or an Ion AmpliSeq Custom panel from Thermo Fisher [3] (discussion section line 419 - 429).
Guo et al. [1] mainly focused on the question whether a MET overexpression detected by IHC can be correlated to a MET gene amplification or MET exon 14 skipping. They conclude that IHC is not a useful method for the detection of genomic changes of the MET locus like gene amplification or MET-Exon 14 skipping. We strictly focused on the amplification of the MET gene comparing the two methods FISH versus NGS. In contrast to [1], we did not correlate these findings to a possible overepression of the MET protein.
Moreover, two institutes performing NGS did use a hybrid capture based assay, not a PCR amplicon-based one as we did in our analysis. Using a hybrid capture based NGS assay, one investigates a larger gene panel and could loose some information of certain gene loci. In contrast to that, amplicon sequencing has a higher on target rate yielding a high specificity and deep coverage.
In contrast to [2] who did a biased, non-consecutive approach selecting n=26 MET gene amplified (FISH; MET GCN >5, Cappuzzo Score) samples of in total n=222 NSCLC cases, we used an unbiased consecutive approach.
In addition, they used a 500+ NGS gene panel (PGDx elioTM) and set a cut off for MET gene amplification of > 3.0 fold change. In our study cases harboring a fold change > 1.6 were classified as MET gene amplified by NGS. For assessment of MET amplification by FISH, they referred to two individual FISH score references (Cappuzzo score and UCCC FISH-criteria). In our study we referred to only one FISH scoring system [4] (discussion section line 430 – 449).
- Additionally, the authors need to correct the references.
We thank the reviewer for mentioning the two new references. Both references have been dicussed in the revised discussion section and the reference section has been updated (reference section line 841 – 846).
- Guo, R.; Berry, L. D.; Aisner, D. L.; Sheren, J.; Boyle, T.; Bunn, P. A., Jr.; Johnson, B. E.; Kwiatkowski, D. J.; Drilon, A.; Sholl, L. M.; Kris, M. G., MET IHC is a Poor Screen for MET Amplification or MET exon 14 mutations in Lung Adenocarcinomas: Data from a Tri-Institutional Cohort of the Lung Cancer Mutation Consortium. J Thorac Oncol 2019.
- S. Clavé, M. S., P. Rocha, M. Hardy-Werbin, J. Gibert, X. Riera, E. Weingartner, G. Cerqueira, D. Nichol, J. Simmons, Á Taus, L. Pijuan, B. Bellosillo, E. Arriola,, 1991P - Identification of MET gene amplifications using next-generation sequencing in non-small cell lung cancer patients,. Annals of Oncology, 2019.
- Heydt, C.; Becher, A.-K.; Wagener-Ryczek, S.; Ball, M.; Schultheis, A. M.; Schallenberg, S.; Rüsseler, V.; Büttner, R.; Merkelbach-Bruse, S., Comparison of in situ and extraction-based methods for the detection of MET amplifications in solid tumors. Computational and Structural Biotechnology Journal 2019, 1339–1347.
- Schildhaus, H. U.; Schultheis, A. M.; Rüschoff, J.; Binot, E.; Merkelbach-Bruse, S.; Fassunke, J.; Schulte, W.; Ko, Y.-D.; Schlesinger, A.; Bos, M.; Gardizi, M.; Engel-Riedel, W.; Brockmann, M.; Serke, M.; Gerigk, U.; Hekmat, K.; Frank, K. F.; Reiser, M.; Schulz, H.; Krüger, S.; Stoelben, E.; Zander, T.; Wolf, J.; Buettner, R., MET Amplification Status in Therapy-Naïve Adeno- and Squamous Cell Carcinomas of the Lung. J Clinical Cancer Research 2015, 21 (4), 907-915.

Round 2
Reviewer 1 Report
The authors have addressed all of my concerns